# An Information Criterion for Controlled Disentanglement of Multimodal Data

**Chenyu Wang**[*,1,2], **Sharut Gupta**[*,1], **Xinyi Zhang**[1,2], **Sana Tonekaboni**[2],

**Stefanie Jegelka**[1,3], **Tommi Jaakkola**[1], **Caroline Uhler**[1,2]

[1]MIT [2]Broad Institute of MIT and Harvard [3]TU Munich

## Abstract

Multimodal representation learning seeks to relate and decompose information available in multiple modalities. By disentangling modality-specific information from information that is shared across modalities, we can improve interpretability and robustness and enable tasks like counterfactual generation. However, separating these components is challenging due to their deep entanglement in real-world data. We propose **Disentangled Self-Supervised Learning** (DISENTANGLEDSSL), a novel self-supervised approach that effectively learns disentangled representations, even when the so-called *Minimum Necessary Information* (MNI) point is not achievable. It outperforms baselines on multiple synthetic and real-world datasets, excelling in downstream tasks, including prediction tasks for vision-language data, and molecule-phenotype retrieval for biological data.

## 1 Introduction

Multimodal representation learning integrates information from different modalities to form holistic representations, with applications ranging from vision-language tasks [40, 21, 25] to biological data analysis [39, 41]. Models like CLIP [25] use self-supervised learning to capture shared information, assuming that mutual information across modalities is key for downstream tasks [34, 31]. However, the modality gap—stemming from inherent differences in representation and content—causes misalignment and limits their effectiveness in real-world scenarios [18, 26, 10]. This highlights the need for a *disentangled representation space* that captures both shared and modality-specific information effectively. Such a space, accounting for both *coverage* and *disentanglement* of the information, is crucial for interpretability and tasks where distinct modalities provide essential insights [42, 17, 20].

Disentangled representation learning in multimodal data began with works on Variational Autoencoders and Generative Adversarial Networks [14, 4, 5] aiming to isolate data variations. Recent self-supervised methods have advanced this by learning shared and modality-specific information either sequentially or jointly [20, 15, 28]. Zhang et al. [42] emphasized the importance of learning disentangled representations for multimodal data in biological contexts. However, rigorous guarantees for disentanglement are lacking, especially when the *Minimum Necessary Information* (MNI) [7] point is unattainable. In many real-world applications, shared and modality-specific information are deeply entangled, i.e. the shared and modality-specific components are intertwined and result in similar observations, leading to unattainable MNI. Figure 1 illustrates an example in the context of high-content drug screens. We provide additional examples of unattainable MNI in Appendix B.

In this work, we propose DISENTANGLEDSSL, a self-supervised approach for multimodal data that effectively separates shared and modality-specific information. Building on information theory, we devise a step-by-step optimization strategy to learn these representations and maximize the variational

---

[*]Equal contribution

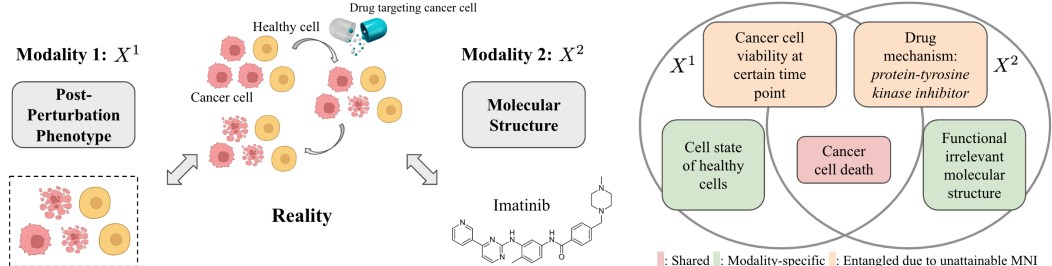

Figure 1: Post-perturbation phenotype ($X_1$) (i.e., cellular images or gene expression after the application of a drug to cells) and molecular structure ($X_2$) of an underlying drug perturbation system where cancer cells are targeted and killed while healthy cells remain unaffected. The Venn diagram illustrates shared and specific information between modalities $X_1$ and $X_2$: shared content is shown in red, modality-specific content in green, and entangled content due to unattainable MNI in orange. For example, for the drug mechanism, the molecular structure conveys full information, while the phenotype provides partial information (i.e. mechanisms causing cell death). Similarly, for the states of healthy cells, the phenotype specifies their cell states, whereas the molecular structure only indicates that the cells are unaffected without detailing their specific states.

lower bound on our objectives. Unlike existing works, we tackle challenging cases where MNI is unattainable and provide a formal analysis of representation optimality. Empirical results show that DISENTANGLEDSSL outperforms baselines on various datasets, including multimodal prediction in MultiBench [16] and molecule-phenotype retrieval in high-content drug screening datasets[32, 3].

## 2 Method

In this section, we detail our proposed method, DISENTANGLEDSSL, for learning disentangled representations in multimodal data. We begin by outlining the graphical model that formalizes the problem in Section 2.1 and defining the key properties of "desirable" representations in Section 2.2. Subsequently, we describe the DISENTANGLEDSSL framework in Section 2.3. We provide theoretical proofs of the optimality in Appendix E, and specific training objectives in Appendix H.

### 2.1 Multimodal Representation Learning with Disentangled Latent Space

DISENTANGLEDSSL learns disentangled representations in latent space, separating modality-specific information from shared factors across paired observations ($X^1$, $X^2$). This generative process is modeled in Figure 2. Each observation is generated from two distinct latent representations: the modality-specific representations ($Z_s^1$ and $Z_s^2$) that contain information exclusive to their respective modalities, and a shared representation ($Z_c$) that contains information common to both modalities. We refer to these as the true latents.

DISENTANGLEDSSL infers the shared representation from both modalities independently, i.e. $\hat{Z}_c^1 \sim p(\cdot|X^1)$ and $\hat{Z}_c^2 \sim p(\cdot|X^2)$. The modality-specific information for each modality is encoded by variables $\hat{Z}_s^1$ and $\hat{Z}_s^2$. Note that for the true latents, $Z_s^1$ and $Z_c$ are conditionally dependent on $X^1$ due to the V-structure in the graphical model. To preserve such dependencies in the inferred latents, $\hat{Z}_s^1$ and $\hat{Z}_s^2$ are conditioned on both the respective observations and the inferred shared representations, with $\hat{Z}_s^1 \sim p(\cdot|X^1, \hat{Z}_c^1)$ and $\hat{Z}_s^2 \sim p(\cdot|X^2, \hat{Z}_c^2)$.

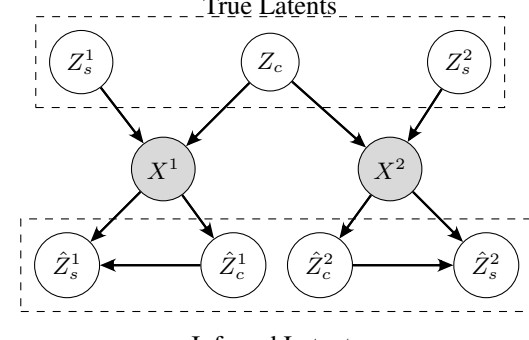

Figure 2: Graphical model.

### 2.2 Information Criteria for the Optimal Inferred Representations

We establish information-theoretic criteria to ensure the shared and modality-specific representations are informative and disentangled, capturing key features while minimizing redundancy.

### 2.2.1 Information Bottleneck Principle and Minimum Necessary Information

The shared representations $\hat{Z}_c^1$ and $\hat{Z}_c^2$ should provide a compact yet expressive form of shared information. This trade-off between compression and expressivity is studied by the principles of the information bottleneck (IB) in both supervised and self-supervised settings [33, 12, 30, 29, 35]. IB objective has been utilized to learn the optimal representations $Z^1$ of $X^1$ with respect to $X^2$ under the Markov chain $Z^1 \leftarrow X^1 \leftrightarrow X^2$. It balances the trade-off between preserving relevant information about $X^2$, i.e. $I(Z^1; X^2)$, and compressing the representation, i.e. $I(Z^1; X^1)$ (see more details in Appendix C). The optimal representation should be both **sufficient**, i.e. $I(Z^1; X^2) = I(X^1; X^2)$ [1, 29], and **minimal** [1]. Based on these criteria, $Z^1$ is said to capture **Minimum Necessary Information (MNI)** [7] between $X^1$ and $X^2$ if $I(X^1; X^2) = I(Z^1; X^2) = I(Z^1; X^1)$, indicating an ideal scenario of full disentanglement between $X^1$ and $X^2$ with no extraneous information, i.e. $I(Z^1; X^1|X^2) = 0^2$.. In general, it may be unattainable for an arbitrary distribution $p(X^1, X^2)$ (see Appendix F). Despite its significance, the optimality of latent representations when MNI is unattainable is often overlooked in prior work.

### 2.2.2 Optimal Shared Representations: MNI attainable or not

We propose a definition of the optimality of the shared representations that applies to both scenarios – when MNI is attainable or not, as defined in Equation 1:

$$\hat{Z}_c^{1*} = \arg\min_{Z^1} I(Z^1; X^1|X^2), \text{ s.t. } I(X^1; X^2) - I(Z^1; X^2) \le \delta_c$$
$$\hat{Z}_c^{2*} = \arg\min_{Z^2} I(Z^2; X^2|X^1), \text{ s.t. } I(X^1; X^2) - I(Z^2; X^1) \le \delta_c \tag{1}$$

Formally, minimizing conditional mutual information, $I(Z^1; X^1|X^2)$, ensures the shared representation captures only the truly common information between $X^1$ and $X^2$, excluding modality-specific details unique to $X^1$. Compared with $I(Z^1; X^1)$ in IB, it provides a more precise measure of compression and a more robust objective.

The constraint $I(X^1; X^2) - I(Z^1; X^2) \le \delta_c$ ensures that $\hat{Z}_c^{1*}$ retains a substantial portion of the shared information between $X^1$ and $X^2$, controlling the difference within the limit $\delta_c$ and preventing significant information loss. We utilize the **IB curve** $F(\delta)^3$ [12, 8], representing the maximum $I(Z^1; X^2)$ for a given compression level $I(X^1; Z^1) \le \delta$, to illustrate the optimality in Figure 3. MNI is depicted as point A, and $\hat{Z}_c^{1*}$ corresponding to $\delta_c$ is shown as point C. When MNI is attainable, setting $\delta_c = 0$ achieves MNI. In contrast, Achille and Soatto [1] formulated the optimization as $Z^1 = \arg\min_{Z^1: Z^1\text{-}X^1\text{-}X^2} I(X^1; Z^1), s.t. I(Z^1; X^2) = I(X^1; X^2)$,

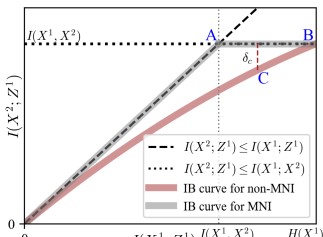

Figure 3: IB Curve

leading to MNI when attainable. This holds in supervised settings, assuming the data label $X^2$ is a deterministic function of $X^1$, as used by previous methods [12, 7, 24]. However, in general multimodal self-supervised scenarios where MNI is not attainable, this results in point B in Figure 3, which includes information of $X^1$ that has little relevance to $X^2$ to satisfy the equality constraint, causing a gap between the objective and the ideal representation.

### 2.2.3 Optimal Specific Representations: Ensuring Coverage and Disentanglement

The modality-specific representations $Z_s^1$ and $Z_s^2$ should capture information unique to each modality, being highly informative yet minimally redundant with the shared representations. The formal definition of these optimal representations is provided in Equation 2.

$$\hat{Z_s^{1*}} = \arg\max_{Z^1} I(Z^1; X^1|X^2), \text{ s.t. } I(Z^1; \hat{Z}_c^{1*}) \le \delta_s$$
$$\hat{Z_s^{2*}} = \arg\max_{Z^2} I(Z^2; X^2|X^1), \text{ s.t. } I(Z^2; \hat{Z}_c^{2*}) \le \delta_s \tag{2}$$

---

$^2 I(Z^1; X^1|X^2) = I(Z^1; X^1, X^2) - I(Z^1; X^2) = I(Z^1; X^1) - I(Z^1; X^2) = 0$, where the first equality is due to the property of conditional mutual information, and the second is due to the Markov structure $Z^1\text{-}X^1\text{-}X^2$.

$^3$The IB curve is concave [8], monotonically non-decreasing, and upper bounded by line $I(Z^1; X^2) = I(Z^1; X^1)$ and line $I(Z^1; X^2) = I(X^1; X^2)$ (see details and proofs in Appendix D).

To extract modality-specific information, $\hat{Z}_s^{1*}$ maximizes the conditional mutual information $I(Z^1; X^1|X^2)$, capturing details unique to $X^1$ without redundancy from $X^2$. This same term is minimized for optimal shared representations in Equation 1. Integrating both representations together, the objectives ensure comprehensive coverage of all relevant information from each modality. Meanwhile, the constraint $I(Z^1; \hat{Z}_c^{1*}) \leq \delta_s$ limits redundancy between the modality-specific representation and the shared representation $\hat{Z}_c^{1*}$, ensuring disentanglement by separating unique aspects of $X^1$ from shared components with $X^2$. The parameter $\delta_s$ controls the trade-off between information coverage and disentanglement.

### 2.3 DISENTANGLEDSSL: a Step-by-Step Optimization Algorithm

To achieve the optimal representations discussed in Section 2.2, we introduce a two-step training procedure. The first step focuses on optimizing for the shared latent representation to captures the minimum necessary information as close as possible. Then, the second step utilizes the learned shared representations in step 1 to facilitate the learning of modality-specific representations. This sequential approach is formalized in the optimization objectives given in Equations 3 and 4.

**Step 1:** Learn the shared latent representations by encouraging the shared representation encoded from one modality to be highly informative about the other modality, while minimizing redundancy.

$$\hat{Z}_c^{1*} = \arg\max_{Z^1} L_c^1 = \arg\max_{Z^1} I(Z^1; X^2) - \beta \cdot I(Z^1; X^1|X^2)$$
$$\hat{Z}_c^{2*} = \arg\max_{Z^2} L_c^2 = \arg\max_{Z^2} I(Z^2; X^1) - \beta \cdot I(Z^2; X^2|X^1)$$

(3)

**Step 2:** Learn the modality-specific latent representations based on the learned shared representations.

$$\hat{Z}_s^{1*} = \arg\max_{Z^1} L_s^1 = \arg\max_{Z^1} I(Z^1, \hat{Z}_c^{2*}; X^1) - \lambda \cdot I(Z^1; \hat{Z}_c^{1*})$$
$$\hat{Z}_s^{2*} = \arg\max_{Z^2} L_s^2 = \arg\max_{Z^2} I(Z^2, \hat{Z}_c^{1*}; X^2) - \lambda \cdot I(Z^2; \hat{Z}_c^{2*})$$

(4)

The hyperparameters $\beta$ and $\delta$ balance relevance and redundancy in the shared and modality-specific representations respectively. We use the same values for both modalities since they operate on similar information scales. Our sequential training approach, instead of a joint one, stems from the self-sufficient nature of each optimization procedure, where a sub-optimal representation does not enhance the learning of the other. We provide theoretical proofs of the optimality of DISENTANGLEDSSL in Appendix E, and specific training objectives for each information term in Appendix H.

## 3 Experimental Results

We conduct experiments on a simulation study (results in Appendix I.1) and two real-world multimodal settings. We compare DISENTANGLEDSSL against a disentangled variational autoencoder baseline, DMVAE [14], as well as various multimodal contrastive learning methods, including CLIP [25], which aligns modalities to learn shared representations, and FOCAL [20] and FactorCL [17], which aim to capture both shared and modality-specific information. Additionally, we test a joint optimization variant, JointOpt [24], to demonstrate the benefits of our step-by-step approach.

### 3.1 MultiBench

We utilize the real-world multimodal benchmark from MultiBench [16], which includes curated datasets across various modalities, such as text, images, and tabular data, along with a downstream label that we expect shared and specific information to have varying importance for. We evaluate the linear probing accuracy of representations learned by each pretrained model, as shown in Table 1. FactorCL-emb refers to the embeddings of FactorCL before projection heads, while FactorCL-proj uses the concatenation of all projection head outputs. All models use representations (or concatenation of them, if applicable) with the same dimensionality. We present results for the combined shared and specific representations for FOCAL and JointOpt in Table 1, with individual results in Appendix I.2. DISENTANGLEDSSL consistently outperforms baselines across all datasets, demonstrating its ability to capture valuable information for downstream tasks. Combining shared and specific representations of DISENTANGLEDSSL improves performance in most cases, showing both

contribute to label prediction, while in MOSI and MUSTARD, only shared or specific information contributes significantly.

Table 1: Prediction accuracy (%) of the representations learned by different methods on MultiBench datasets and standard deviations over 3 random seeds.

| Dataset | MIMIC | MOSEI | MOSI | UR-FUNNY | MUSTARD |
|---|---|---|---|---|---|
| CLIP | 64.97 (0.60) | 76.87 (0.45) | 64.24 (0.88) | 62.73 (0.92) | 56.04 (4.19) |
| FactorCL-emb | 65.25 (0.45) | 71.80 (0.64) | 62.97 (0.81) | 63.29 (2.07) | 56.76 (4.66) |
| FactorCL-proj | 59.43 (1.70) | 74.61 (1.65) | 56.02 (1.26) | 61.25 (0.47) | 55.80 (2.18) |
| FOCAL | 64.42 (0.34) | 76.77 (0.51) | 63.65 (1.09) | 62.98 (1.52) | 54.35 (0.00) |
| JointOpt | 66.11 (0.64) | 76.71 (0.14) | 64.24 (1.75) | 63.58 (1.45) | 56.52 (2.61) |
| DISENTANGLEDSSL (shared) | 63.16 (0.48) | 76.94 (0.22) | **65.16** (0.81) | 64.14 (1.53) | 54.11 (1.51) |
| DISENTANGLEDSSL (specific) | 65.73 (0.09) | 75.99 (0.60) | 51.70 (0.72) | 60.27 (1.28) | **61.60** (2.61) |
| DISENTANGLEDSSL (both) | **66.44** (0.31) | **77.45** (0.06) | 65.11 (0.80) | 64.24 (1.54) | 56.52 (2.18) |

## 3.2 High-content Drug Screening.

**Dataset description.** As characterized in Figure 1, we use two high-content drug screening datasets which provide phenotypic profiles after drug perturbation: RXRX19a [32] containing cell imaging profiles, and LINCS [3]. We conduct train-validation-test splitting according to molecules. Models are pretrained to learn representations of molecular structures and corresponding phenotypes. We provide additional details on experimental settings in Appendix I.3.

**Molecule-phenotype retrieval using shared representations.** We evaluate the shared representations on the molecule-phenotype retrieval task, which identifies molecules likely to induce a specific phenotype. The shared information, which connects the molecular structure and phenotype, plays a key role in this task. We tune $\beta$ according to validation set performance and show results of top N accuracy (N=1,5,10) and mean reciprocal rank (MRR) on the test set in Table 2. DISENTANGLEDSSL significantly outperforms baselines on both datasets, demonstrating its effectiveness in capturing relevant shared features while excluding irrelevant details. Notably, compared to the variant without the information bottleneck constraint, i.e. DISENTANGLEDSSL ($\beta = 0$), the full DISENTANGLEDSSL model preserves critical shared features, achieving superior performance in this retrieval task, where shared information is essential.

Table 2: Retrieving acccuracy and mean reciprocal rank (MRR) of molecule-phenotype retrieval.

| Dataset | RXRX19a | | | | LINCS | | | |
|---|---|---|---|---|---|---|---|---|
| Top N Acc (%) | N=1 | N=5 | N=10 | MRR | N=1 | N=5 | N=10 | MRR |
| Random | 0.30 | 1.50 | 3.00 | - | 0.15 | 0.75 | 1.51 | - |
| CLIP | 3.30(0.40) | 8.33(0.52) | 11.59(0.20) | 0.103(0.001) | 3.95(0.04) | 10.81(0.06) | 15.10(0.20) | 0.146(0.001) |
| DMVAE | 3.85(0.36) | 8.76(0.30) | 11.84(0.32) | 0.106(0.002) | 4.31(0.09) | 11.45(0.13) | 15.88(0.17) | 0.156(0.001) |
| JointOpt | 3.41(0.49) | 8.54(0.14) | 11.64(0.14) | 0.110(0.002) | **4.67**(0.09) | 11.60(0.11) | 16.02(0.15) | 0.161(0.001) |
| FOCAL | 3.61(0.51) | 8.71(0.69) | 11.94(0.74) | 0.108(0.003) | 4.34(0.17) | 11.24(0.19) | 15.74(0.26) | 0.157(0.001) |
| DISENTANGLEDSSL ($\beta = 0$) | 3.39(0.54) | 8.25(0.33) | 11.53(0.20) | 0.109(0.003) | 4.36(0.13) | 11.27(0.39) | 15.81(0.47) | 0.158(0.001) |
| DISENTANGLEDSSL | **4.03**(0.39) | **9.62**(0.20) | **13.12**(0.23) | **0.111**(0.001) | 4.48(0.21) | **11.70**(0.40) | **16.39**(0.39) | **0.163**(0.002) |

**Disentanglement measurement.** To assess the effectiveness of learning modality-specific representations, we introduce the Reconstruction Gain (RG) metric, which quantifies the disentanglement between shared and modality-specific representations. We use 2-layer MLP decoders to reconstruct the original data from the shared and modality-specific representations, both individually and jointly, and compute the reconstruction $R^2$ for each case. A higher gain in $R^2$ when using combined representations indicates lower redundancy and better disentanglement.

As shown in Table 3, DISENTANGLEDSSL achieves the highest RG scores across both modalities and datasets, demonstrating superior disentanglement. In contrast, other methods show redundancy due to insufficient constraints for disentanglement during pretraining.

Table 3: Reconstruction gain (RG) in $R^2$ of representations for each modality (molecular structure/phenotype).

| Dataset | RXRX19a | | LINCS | |
|---|---|---|---|---|
| Metric | RG-molecule | RG-phenotype | RG-molecule | RG-phenotype |
| FOCAL | 0.117(0.006) | 0.547(0.001) | 0.122(0.001) | 0.618(0.002) |
| DMVAE | 0.123(0.002) | 0.545(0.003) | 0.139(0.002) | 0.605(0.003) |
| JointOpt | 0.130(0.001) | 0.524(0.001) | 0.103(0.000) | 0.604(0.002) |
| DISENTANGLEDSSL | **0.153**(0.003) | **0.591**(0.001) | **0.143**(0.000) | **0.622**(0.002) |

We also test the representations from the pretrained model on the counterfactual generation task, with detailed results provided in Appendix I.3.

## Acknowledgement

Xinyi Zhang, Sana Tonekaboni and Caroline Uhler were partially supported by the Eric and Wendy Schmidt Center at the Broad Institute, NCCIH/NIH (1DP2AT012345), ONR (N00014-22-1-2116) and DOE (DE-SC0023187). SG and SJ acknowledge the support of the NSF Award CCF-2112665 (TILOS AI Institute) and an Alexander von Humboldt fellowship. CW and TJ acknowledge support from the Machine Learning for Pharmaceutical Discovery and Synthesis (MLPDS) consortium, the DTRA Discovery of Medical Countermeasures Against New and Emerging (DOMANE) threats program, and the NSF Expeditions grant (award 1918839) Understanding the World Through Code.

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

## A   Related Work

**Disentangled representations in VAEs and GANs.** Disentangled representation learning originated from works on Variational Autoencoders (VAEs) and Generative Adversarial Networks (GANs), focusing on isolating underlying data variations linked to or independent of labels (e.g., digit identity vs. writing style in MNIST) [9, 22, 2, 4]. This concept extended to multimodal data like text and image [13] and content and pose in video [5], typically using self- and cross-reconstruction with adversarial loss to learn shared and specific components. However, these methods often address simple cases with attainable MNI, lacking a comprehensive analysis for complex real-world multimodal scenarios where MNI is not attainable.

**Information bottleneck and its variants.** The information bottleneck (IB) principle has been used to analyze and optimize deep neural networks and the learned representations from the information theory perspective in both supervised and self-supervised settings [33, 8, 30, 12, 27, 35, 38, 29], with extensions like the conditional entropy bottleneck (CEB) [7, 13] being developed to better refine the information plane. IB has been applied to learn molecular representations from drug screening data [19]. However, these approaches primarily focus on extracting shared features while minimizing specific features, limiting their practical use. Other methods [24, 28] learn disentangled representations by optimizing mutual information and using adversarial loss but are confined to single-modality scenarios and do not address complex multimodal settings with unattainable MNI.

**Multimodal disentanglement in self-supervised learning.** Recent studies have explored disentangled representation learning in multimodal self-supervised contexts. Liang et al. [17] factorizes and optimizes mutual information bounds to capture the union of shared and unique features, while not penalizing for redundancy between latents. Zhang et al. [42] learns disentangled representations of cell state for multimodal data in biological contexts. Liu et al. [20] and Li et al. [15] use mutual information optimization with orthogonal/MMD regularizations for disentangling multimodal time-series signals and T-cell receptor segments, respectively. However, these methods lack a general framework for understanding information content, particularly when data modalities are deeply entangled in real-world applications.

## B   Additional Examples of Unattainable MNI

In this section, we present two additional examples where the Minimum Necessary Information (MNI) is unattainable. These examples offer a clear understanding of how unattainable MNI manifests in data, providing practical insights and making the concept more accessible.

Firstly, we show an example in a simple mathematical case. As illustrated in Figure 4, we have true latent variables are independently drawn from a Bernoulli distribution, i.e. $Z_s^1, Z_c, Z_s^2 \sim$ Bernoulli$(0.5)$, and observations $X^1$ and $X^2$ are generated according to $X^1 = \text{OR}(Z_c, Z_s^1)$ and $X^2 = \text{OR}(Z_c, Z_s^2)$. In this scenario, if we observe $X^1 = 1$, we are not able to distinguish whether this result is due to $Z_c = 1$, or $Z_s^1 = 1$, or both. Thus, extracting purely shared features from the observations is infeasible.

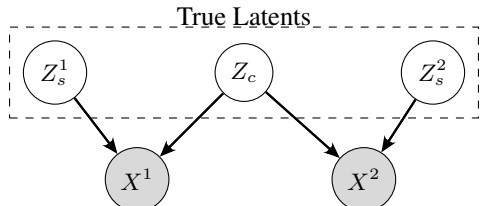

Figure 4: Example of unattainble MNI.

Further, we provide an example in 3D geometry, as illustrated in Figure 5. For information such as the height of the cone and the number of spheres, one modality conveys the complete information while the other conveys partial information. This ambiguity makes it challenging to classify the information as purely shared or modality-specific, placing it in an uncertain category.

## C   Background on Information Bottleneck

The Information Bottleneck (IB) principle provides a powerful theoretical foundation for our method. IB objective seeks to find a representation $Z^1$ of a random variable $X^1$ that optimally trades off the preservation of relevant information with compression [33]. Relevance is defined as the mutual information, $I(Z^1; X^2)$, between the representation $Z^1$ and another target variable $X^2$. Compression

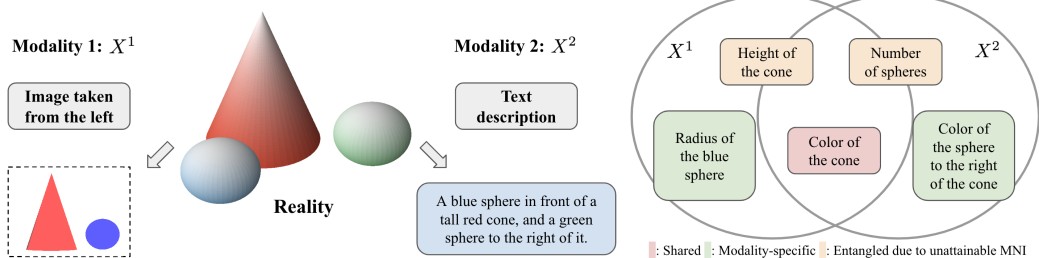

Figure 5: Image modality ($X_1$) and text modality ($X_2$) of an underlying reality. The Venn diagram illustrates shared and specific information between modalities $X_1$ and $X_2$: shared content is shown in red, modality-specific content in green, and entangled content due to unattainable MNI in orange. For example, for the cone's height, the image conveys full information, while the text (i.e. "tall red cone") provides partial information. Similarly, for the number of spheres, the text specifies it completely, whereas the image indicates there is at least one sphere.

is enforced by constraining the mutual information between the representation and the original data, $I(X^1; Z^1)$, to lie below a specified threshold. This is can be formalize in terms of the following constrained optimization problem:

$$\arg\max_{Z^1} I(Z^1; X^2), s.t. \ I(Z^1; X^1) \leq \delta \tag{5}$$

where $Z^1$ is from a set of random variables that obey the Markov chain $Z^1 \leftrightarrow X^1 \leftrightarrow X^2$. In practice, Equation 5 can be optimized by minimizing the IB Lagrangian in Equation 6.

$$\mathcal{L} = I(Z^1; X^1) - \beta I(Z^1; Z^2) \tag{6}$$

Here, the Lagrangian multiplier $\beta$ controls the emphasis placed on compression versus expressiveness.

Fischer [7] extends IB to the conditional entropy bottleneck (CEB) objective, which utilizes the conditional mutual information term $I(Z^1; X^1 | X^2)$ in place of the mutual information term $I(Z^1; X^1)$ in Equation 6:

$$\mathcal{L} = I(Z^1; X^1 | X^2) - \beta I(Z^1; Z^2)$$

While being equivalent to the IB Lagrangian in terms of the information criterion, CEB rectifies the information plane to precisely measure compression using the conditional mutual information, and offers a clear assessment of how close the compression is to being optimal.

Note that although CEB aligns with our step 1 objective in Equation 3, its optimality has not been fully examined in Fischer [7], particularly under scenarios where MNI is unattainable. Furthermore, Fischer [7] primarily addresses a supervised learning context, where $X^2$ serves as the label for $X^1$. In contrast, our study tackles the more complex multimodal setting, where $X^1$ and $X^2$ are two distinct data modalities. We extend the analysis to both attainable and unattainable MNI cases, demonstrating the efficacy of our approach in capturing shared and modality-specific information under these challenging scenarios. This broadens the applicability of CEB beyond traditional label-based supervised settings to multimodal data with complex modality entanglements.

## D   Properties of the IB curve

The **information plane** is a helpful visualization of the information bottleneck principle, which utilizes $I(X^1; Z^1)$ and $I(X^2; Z^1)$ as its coordinates that represents the trade-off between compression and prediction [33]. The frontier of all possible $Z^1$s is called the **IB curve** or $F(r)$ [12] as follows:

$$F(r) := \max_{Z^1: Z^1 - X^1 - X^2} I(Z^1; X^2) \text{ s.t. } I(X^2; Z^1) \leq r$$

The IB curve has the following properties:

- The IB curve is concave (Lemma 5 in [8]) and monotonically non-decreasing.

- The IB curve is upper bounded by line $I(X^2; Z^1) = I(X^1; Z^1)$ and line $I(X^2; Z^1) = I(X^1; X^2)$, according to the Markov relationship $Z^1 - X^1 - X^2$.

- When the MNI point is attainable for random variables $X^1$ and $X^2$, the IB curve has the following formula [27, 12]:

$$I(Z^1; X^2) = \begin{cases} I(Z^1; X^1), & \text{if } I(Z^1; X^1) \leq I(X^1; X^2) \\ I(X^1; X^2), & \text{if } I(X^1; X^2) < I(Z^1; X^1) \leq H(X^1) \end{cases}$$

and the Gateaux derivative of $I(X^2; Z^1)$ with respect to $p(z^1|x^1)$ (as used in [24]) doesn't exist at the MNI point (see details in Appendix G).

# E    Optimality Guarantee for the Learned Representations

## E.1    Optimality Guarantee for the Learned Shared Representations

This section explores how the step 1 objective, $L_c^1$, optimizes the shared representation between modalities by balancing expressivity and redundancy. We discuss its effectiveness in both scenarios–when MNI is attainable or not.

The step 1 objective $L_c^1$ in Equation 3 seeks representation $Z^1$ that maximizes the information shared between the modalities, i.e. $I(Z^1; X^2)$, while minimizing the information unique to each modality, i.e. $I(Z^1; X^1|X^2)$, to capture only the essential shared content. This aligns with the conditional entropy bottleneck (CEB) objective [7], an extension of the IB Lagrangian $L = I(Z^1; X^2) - \tilde{\beta} I(Z^1; X^1)$ (see details in Appendix C). While it serves as a robust objective for learning shared information, its optimality remains underexplored in Fischer [7], particularly when MNI is unattainable. Additionally, Fischer [7] focuses on the supervised scenario where $X^2$ is the label of $X^1$, whereas we address the multimodal case with $X^1$ and $X^2$ being two data modalities, demonstrating its effectiveness in both attainable and unattainable MNI scenarios.

**When MNI is attainable**, Proposition 1 states that the step 1 optimization achieves MNI for any positive $\beta$.

**Proposition 1.** *If MNI is attainable for random variable $X^1$ and $X^2$, maximizing $L_c^1 = I(Z^1; X^2) - \beta I(Z^1; X^1|X^2)$ achieves MNI for any $\beta > 0$, i.e. $I(\hat{Z}_c^{1*}; X^1) = I(\hat{Z}_c^{1*}; X^2) = I(X^1; X^2)$, where $\hat{Z}_c^{1*} := \arg\max_{Z^1 - X^1 - X^2} L_c^1$.*

*Proof.* Based on data processing inequality and the Markov relationship $\hat{Z}_c^1 \leftarrow X^1 \leftrightarrow X^2$, $I(\hat{Z}_c^1; X^2) \leq I(X^1; X^2)$. Based on the non-negativity of conditional mutual information, $I(\hat{Z}_c^1; X^1|X^2) \geq 0$. Thus, for $\beta > 0$,

$$L_c^1 = I(\hat{Z}_c^1; X^2) - \beta \cdot I(\hat{Z}_c^1; X^1|X^2) \leq I(X^1; X^2), \forall \hat{Z}_c^1$$

where the equality holds when $I(\hat{Z}_c^1; X^2) = I(X^1; X^2)$ and $I(\hat{Z}_c^1; X^1|X^2) = 0$

Meanwhile,

$$I(\hat{Z}_c^1; X^1|X^2) = I(\hat{Z}_c^1; X^1, X^2) - I(\hat{Z}_c^1; X^2) = I(\hat{Z}_c^1; X^1) - I(\hat{Z}_c^1; X^2)$$

where the second equality is due to the conditional independence $\hat{Z}_c^1 \perp\!\!\!\perp X^2|X^1$ in the Markov relationship.

Therefore, $L_c^1$ achieves maximality $I(X^1; X^2)$ iff $I(\hat{Z}_c^1; X^2) = I(X^1; X^2) = I(\hat{Z}_c^1; X^1)$, i.e. $\hat{Z}_c^1$ achieves MNI.    $\square$

**When MNI is unattainable**, for any representation $Z^2$ of $X^1$, the inequalities $I(Z^2; X^1) \geq I(Z^2; X^2)$ and $I(Z^2; X^1) \geq I(X^1; X^2)$ can not achieve equality simultaneously. This indicates the existence of an inherent trade-off between capturing the entire shared information and avoiding the inclusion of modality-specific details. More precisely, under the condition of strict concavity of the $I(Z^1; X^2) - I(Z^1; X^1)$ information curve[4], such trade-off is presented in Proposition 2.

---

[4]When the IB curve is not strictly concave (i.e. partially linear), the properties hold except for the linear sections. In these cases, modifications like squared IB [12] can be used to achieve bijection mapping.

**Proposition 2.** *For random variables $X^1$ and $X^2$, when the IB curve $I(Z^1; X^2) = F(I(Z^1; X^1))$ is strictly concave,*
*1) there exists a bijective mapping from $\beta$ in $L_c^1$ to the value of information constraint $\delta_c$ in the definition of optimal shared latent $\hat{Z}_c^{1*}$ in Equation 1;*
*2) $\frac{\partial I(Z_\beta^{1*}; X^1)}{\partial \beta} < 0$, $\frac{\partial I(Z_\beta^{1*}; X^2)}{\partial \beta} < 0$, where $Z_\beta^{1*}$ is the optimal solution corresponding to a certain $\beta$.*

*Proof.* Since the IB curve $I(Z^1; X^2) = f_{IB}(I(Z^1; X^1))$ is monotonically non-decreasing and strictly concave, it is monotonically increasing. When $Z^1 = X^1$, $f_{IB}$ achieves the maximum point $(H(X^1), I(X^1; X^2))$. Thus, $I(X^1; X^2) \geq I(X^1; X^2) - I(\hat{Z}_c^{1*}; X^2) \geq 0$ and is monotonically decreasing. Then there is a bijective mapping between $\delta = I(X^1; X^2) - I(\hat{Z}_c^{1*}; X^2) \in [0, I(X^1; X^2)]$ and points in the IB curve.

Since $I(Z^1; X^1|X^2) = I(Z^1; X^1) - I(Z^1; X^2)$, we have $I(Z^1; X^1|X^2) = f_{IB}^{-1}(I(Z^1; X^2)) - I(Z^1; X^2) := f_{CIB}^{-1}(I(Z^1; X^2))$, where $f_{CIB}(r) = (f_{IB}^{-1}(r) - r)^{-1}$. Since $f_{IB}(r) \leq r$, the equality holds when $r = 0$, and $f_{IB}$ is strictly concave, we have $\frac{df_{IB}}{dr} \leq \frac{df_{IB}}{dr}|_{r=0} < 1$. Thus $\frac{df_{IB}^{-1}}{dr} > 1$. Then $\frac{df_{CIB}}{dr} = (\frac{df_{IB}^{-1}}{dr} - 1)^{-1} > 0$. Furthermore, since $\frac{df_{IB}^2}{dr^2} < 0$ (strict concavity of $f_{IB}$), $\frac{df_{CIB}^2}{dr^2} = -\frac{1}{(\frac{df_{IB}^{-1}}{dr} - 1)^2} \cdot (-\frac{\frac{df_{IB}^2}{dr^2}}{(\frac{df_{IB}^{-1}}{dr})^2}) < 0$. Therefore, $f_{CIB}$ is also monotonically increasing and strictly concave. Then there is a bijective mapping between $\delta = I(X^1; X^2) - I(\hat{Z}_c^{1*}; X^2) \in [0, I(X^1; X^2)]$ and points in the CIB curve $f_{CIB}$.

Since $f_{CIB}$ is increasing, the inequality constraint can be replaced by the equality constraint, i.e.

$$\hat{Z}_c^{1*} = \underset{I(Z^1; X^2) \geq I(X^1; X^2) - \delta}{\arg\min} I(Z^1; X^1|X^2) = \underset{I(Z^1; X^2) = I(X^1; X^2) - \delta}{\arg\min} I(Z^1; X^1|X^2)$$

The corresponding Lagrangian function is

$$L = I(Z^1; X^1|X^2) - \tilde{\beta} \cdot (I(Z^1; X^2) - I(X^1; X^2) + \delta) \tag{7}$$
$$= f_{CIB}^{-1}(I(Z^1; X^2)) - \tilde{\beta} \cdot I(Z^1; X^2) + \tilde{\beta} \cdot (I(X^1; X^2) - \delta)$$

Based on the Lagrangian multiplier theorem, the optimal $\hat{Z}_c^{1*}$ is achieved when

$$\frac{dL}{dI(Z^1; X^2)} = \frac{df_{CIB}^{-1}(I(Z^1; X^2))}{dI(Z^1; X^2)} - \tilde{\beta} = 0$$

$$\frac{dL}{d\tilde{\beta}} = -I(Z^1; X^2) + I(X^1; X^2) - \delta = 0$$

Thus,

$$\tilde{\beta} = \frac{df_{CIB}^{-1}(I(Z^1; X^2))}{dI(Z^1; X^2)}|_{I(Z^1; X^2) = I(X^1; X^2) - \delta} \tag{8}$$

i.e. $\tilde{\beta}$ is the slope of $f_{CIB}^{-1}$ when $I(Z^1; X^2) = I(X^1; X^2) - \delta$. Therefore, there is a bijective mapping between $\beta = \tilde{\beta}^{-1}$ in $L_c^1$ and points in $f_{CIB}$, and thus $\delta$.

For any $\beta_1, \beta_2 > 0$ and $\beta_1 > \beta_2$, we have $\tilde{\beta}_1 = \beta_1^{-1} < \beta_2^{-1} = \tilde{\beta}_2$. According to formula 8,

$$\tilde{\beta}_1 = \frac{df_{CIB}^{-1}(I(Z^1; X^2))}{dI(Z^1; X^2)}|_{I(Z^1; X^2) = I(Z_{\beta_1}^*; X^2)}, \quad \tilde{\beta}_2 = \frac{df_{CIB}^{-1}(I(Z^1; X^2))}{dI(Z^1; X^2)}|_{I(Z^1; X^2) = I(Z_{\beta_2}^*; X^2)} \tag{9}$$

Since $\tilde{\beta}_1 < \tilde{\beta}_2$ and $f_{CIB}$ is a strictly concave function, $f_{CIB}^{-1}$ is strictly convex and we have $I(Z_{\beta_1}^*; X^2) < I(Z_{\beta_2}^*; X^2)$. Furthermore, since $f_{IB}$ is monotonically increasing, we have $I(Z_{\beta_1}^*; X^1) < I(Z_{\beta_2}^*; X^1)$. Therefore,

$$\frac{\partial I(Z_\beta^*; X^2)}{\partial \beta} < 0, \quad \frac{\partial I(Z_\beta^*; X^1)}{\partial \beta} < 0 \tag{10}$$

$\square$

The second property implies that as $\beta$ increases, the learned representation becomes less informative and redundant, demonstrating the tradeoff between expressivity and redundancy when MNI is unattainable. Furthermore, the bijection mapping between $\beta$ and $\delta_c$ allows our step 1 optimization to navigate the information frontier across various $\beta$ values, facilitating soft control and the segmentation of shared information into different degrees of granularity.

### E.2 Optimality Guarantee for the Learned Modality-Specific Representations

In this section, we demonstrate the step 2 objective, $L_s^1$, ensures optimal coverage and disentanglement by showing its equivalence (or nearly equivalence) to the Lagrangian of Equation 2.

The step 2 objective $L_s^1$ in Equation 4 learns modality-specific representation $Z^1$ based on the optimal shared representations $\hat{Z}_c^{1*}$ and $\hat{Z}_c^{2*}$ learned from step 1. It aims to maximize the information coverage of the data observation $X^1$ through the combination of $Z^1$ and $\hat{Z}_c^{2*}$, i.e. $I(Z^1, \hat{Z}_c^{2*}; X^1)$. Simultaneously, it promotes disentanglement by limiting overlap with the shared representation $\hat{Z}_c^{1*}$ of the same modality, indicated by $I(Z^1; \hat{Z}_c^{1*})$. Formally, $L_s^1$ is the Lagrangian formulation of the constraint optimization for optimal modality-specific representations, as defined in Equation 2, however having the term $I(Z^1; X^1 | X^2)$ substituted with $I(Z^1, \hat{Z}_c^{2*}; X^1)$.

**When MNI is attainable**, as justified in Proposition 3, this substitution results in an equivalent objective. Consequently, $L_s^1$ effectively guides the generation of optimal modality-specific representations.

**Proposition 3.** *If MNI is attainable for random variables $X^1$ and $X^2$,*

$$\underset{Z^1 - X^1 - X^2}{\arg\max} \; I(Z^1; X^1 | X^2) = \underset{Z^1 - X^1 - X^2}{\arg\max} \; I(Z^1, \hat{Z}_c^{2*}; X^1)$$

*where $\hat{Z}_c^{2*}$ is the representation based on $X^2$ that satisfies MNI, i.e. $I(\hat{Z}_c^{2*}; X^1) = I(\hat{Z}_c^{2*}; X^2) = I(X^1; X^2)$.*

*Proof.* Based on the graphical model, $\hat{Z}_c^2 \perp\!\!\!\perp X^1 | X^2, Z^1$, we have $I(\hat{Z}_c^2; X^1 | Z^1, X^2) = 0$. Thus,

$$I(Z^1, X^2; X^1) = I(Z^1, X^2; X^1) + I(\hat{Z}_c^2; X^1 | Z^1, X^2) \tag{11}$$
$$= I(Z^1, X^2, \hat{Z}_c^2; X^1) = I(Z^1, \hat{Z}_c^2; X^1) + I(X^1; X^2 | Z^1, \hat{Z}_c^2)$$

Meanwhile,

$$I(X^1; X^2 | \hat{Z}_c^2) - I(X^1; X^2 | Z^1, \hat{Z}_c^2) = I(X^2; Z^1 | \hat{Z}_c^2) - I(X^2; Z^1 | X^1, \hat{Z}_c^2) = I(X^2; Z^1 | \hat{Z}_c^2)$$

where the second equality is due to the conditional independence $X^2 \perp\!\!\!\perp Z^1 | X^1, \hat{Z}_c^2$ in the graphical model.

When $\hat{Z}_c^2$ is the MNI point $\hat{Z}_c^{2*}$, $I(X^1; \hat{Z}_c^{2*}) = I(X^1; X^2)$. Thus,

$$I(X^1; X^2 | \hat{Z}_c^{2*}) = I(X^1; X^2, \hat{Z}_c^{2*}) - I(X^1; \hat{Z}_c^{2*}) = I(X^1; X^2) - I(X^1; \hat{Z}_c^{2*}) = 0 \tag{12}$$

where the first equality is due to the Markov relationship $X^1 \leftrightarrow X^2 \rightarrow \hat{Z}_c^{2*}$.

Then we have $-I(X^1; X^2 | Z^1, \hat{Z}_c^{2*}) = I(X^2; Z^1 | \hat{Z}_c^{2*})$. Due to the non-negativity of conditional mutual information,

$$I(X^1; X^2 | Z^1, \hat{Z}_c^{2*}) = I(X^2; Z^1 | \hat{Z}_c^{2*}) = 0$$

Based on formula 11 and 13,

$$I(Z^1, X^2; X^1) = I(Z^1, \hat{Z}_c^{2*}; X^1)$$

Since $I(Z^1; X^1 | X^2) = I(Z^1, X^2; X^1) - I(X^1; X^2)$ and $I(X^1; X^2)$ is a constant value irrelevant to $Z^1$, maximizing $I(Z^1; X^1 | X^2)$ is equivalent to maximizing $I(Z^1, X^2; X^1) = I(Z^1, \hat{Z}_c^{2*}; X^1)$.

$\square$

**When MNI is unattainable**, as established in Proposition 4, such substitution yields an objective that is nearly equivalent, subject to the value of $\delta_c$ that corresponds to the $\beta$ used in step 1 optimization. Therefore, optimizing $L_s^1$ leads to nearly optimal modality-specific representations.

**Proposition 4.** *For random variables $X^1$ and $X^2$,*

$$0 \le I(Z^1, X^2; X^1) - I(Z^1, \hat{Z}_c^{2*}; X^1) \le \delta_c$$

*where $\hat{Z}_c^{2*}$ is the optimal representation based on $X^2$ with respect to $\delta_c$ as defined in Equation 1, i.e. $\hat{Z}_c^{2*} = \arg\min_{Z^2} I(Z^2; X^2 | X^1),\ s.t.\ I(X^1; X^2) - I(Z^2; X^1) \le \delta_c$.*

*Proof.* According to formula 11 and 12,

$$I(Z^1, X^2; X^1) = I(Z^1, \hat{Z}_c^2; X^1) + I(X^1; X^2 | Z^1, \hat{Z}_c^2)$$

$$I(X^1; X^2 | Z^1, \hat{Z}_c^2) = I(X^1; X^2 | \hat{Z}_c^2) - I(Z^1; X^2 | \hat{Z}_c^2)$$

Since $\hat{Z}_c^{2*}$ is the optimal latent under $\delta$, $I(X^1; X^2) - I(\hat{Z}_c^{2*}; X^1) \le \delta$. Thus,

$$I(X^1; X^2 | \hat{Z}_c^{2*}) = I(X^1; X^2) - I(X^1; \hat{Z}_c^{2*}) \le \delta$$

Meanwhile, $I(Z^1; X^2 | \hat{Z}_c^2) \ge 0$. Then we have $I(X^1; X^2 | Z^1, \hat{Z}_c^2) \le \delta - I(Z^1; X^2 | \hat{Z}_c^2) \le \delta$. Therefore,

$$0 \le I(Z^1, X^2; X^1) - I(Z^1, \hat{Z}_c^2; X^1) = I(X^1; X^2 | Z^1, \hat{Z}_c^2) \le \delta$$

i.e. $|I(Z^1, X^2; X^1) - I(Z^1, \hat{Z}_c^2; X^1)| \le \delta$

Since $I(Z^1; X^1 | X^2) = I(Z^1, X^2; X^1) - I(X^1; X^2)$ and $I(X^1; X^2)$ is a constant value irrelevant to $Z^1$, maximizing $I(Z^1; X^1 | X^2)$ is equivalent to maximizing $I(Z^1, X^2; X^1) \in [I(Z^1, \hat{Z}_c^{2*}; X^1), I(Z^1, \hat{Z}_c^{2*}; X^1) + \delta]$.

$\square$

# F  Sufficient Conditions for MNI

In this section, we explore the conditions for both MNI attainable and unattainable cases. Here, we use $X$ and $Y$ to represent the two modalities, rather than $X^1$ and $X^2$ in other sections. Although prior works have not provided a comprehensive necessary and sufficient condition for the attainability of MNI, and such analysis is beyond the scope of this paper, we provide several sufficient conditions for both MNI attainable and unattainable scenarios.

We first present the sufficient conditions under which MNI is attainable. As outlined in Proposition 5, Proposition 6, and Proposition 7, MNI is attainable when the relationships between $X$ and $Y$ are either entirely deterministic or completely independent across each sub-domain. While the deterministic mapping might hold true when $Y$ is the data label, it generally does not hold when $X$ and $Y$ are high-dimensional data modalities. In such cases, a deterministic relationship between the two modalities would imply that one can be fully inferred from the other, leaving no room for modality-specific information in one of the modalities.

**Proposition 5.** *For random variables $X$, $Y$, and representation $Z$ derived from $X$ (i.e., the Markov chain $Z \leftarrow X \leftrightarrow Y$ holds), a sufficient condition for MNI to be attainable is that $X \to Y$ mapping is deterministic [7].*

*Proof.* Denote $Y = f(X)$ as the deterministic $X \to Y$ mapping. If the encoder is powerful enough, it can learn to reproduce the deterministic function $f$, i.e. $Z = f(X) = Y$. Thus $I(X; Z) = I(X; Y) = H(Y) = I(Y; Z)$. $\square$

While Fischer [7] identifies this as a necessary condition for attainable MNI, there are scenarios where $X \to Y$ is not deterministic, yet MNI is still attainable. We provide additional sufficient conditions for these more general cases in Proposition 6 and Proposition 7.

**Proposition 6.** *For random variables $X$, $Y$, and representation $Z$ derived from $X$ (i.e., the Markov chain $Z \leftarrow X \leftrightarrow Y$ holds), a sufficient condition for MNI to be attainable is that $Y \to X$ mapping is deterministic.*

*Proof.* Denote $X = f(Y)$ as the deterministic $Y \to X$ mapping. Then, for any $Z$ with $Z \leftarrow X \leftrightarrow Y$,

$$p(z|y) = \int_x p(z|x)p(x|y)dx = \int_x p(z|x)\mathbf{1}_{x=f(y)}dx = p(z|x = f(y))$$

More formally, $p(z|y) = p(z|X = f(y))$, and

$$p(Y = y, Z = z) = p(Y = y, X = f(y), Z = z) = p(Y = y) \cdot p(z|X = f(y))$$

Thus the conditional entropy can be written as

$$H(Z|Y) = -\int_{Y,Z} p(y,z) \log p(z|y) dy dz = -\int_Y p(y) \int_Z p(z|X = f(y)) \log p(z|X = f(y)) dy dz$$

$$H(Z|X) = -\int_X p(x) \int_Z p(z|x) \log p(z|x) dx dz$$

Thus, $H(Z|Y) = H(Z|X)$, i.e. $I(Y;Z) = I(X;Z)$. By selecting $Z$ such that $I(Z;X) = I(X;Y) = H(X)$, we achieve the MNI point. $\square$

**Proposition 7.** *For random variable $X, Y$ in $\mathcal{X} \times \mathcal{Y}$ with joint distribution $p(X, Y)$, a sufficient condition for the existence of MNI is:*

*In any sub-domain $\mathcal{X}_s \times \mathcal{Y}_s$ of $\mathcal{X} \times \mathcal{Y}$ where $p(X, Y)$ has the full support, i.e. $\forall\, x, y \in \mathcal{X}_s \times \mathcal{Y}_s, p(x, y) > 0$, $X$ and $Y$ are independent, i.e. $X \perp\!\!\!\perp Y | \{(X, Y) \in \mathcal{X}_s \times \mathcal{Y}_s\}$.*

*Proof.* Construct the learned representation $Z$ as the conditional probability of $Y$ given $X$, i.e. $Z = p(Y|X)$. $Z$ is a deterministic function based o $X$. We will show that such $Z$ satisfies the MNI condition.

First, we show that $I(Z;Y) = I(X;Y)$. Denote $z = p(Y|X = x)$. Since $z$ fully describes the conditional distribution $p(Y|X = x)$, we have $p(Y|X = x) = p(Y|Z = z)$. Thus, $H(Y|X = x) = H(Y|Z = z)$. Therefore,

$$H(Y|X) = \int p(x)H(Y|X = x)dx = \int p(x)H(Y|Z = z)dx$$

$$= \int \left( \int_{\{x:p(Y|X=x)=z\}} p(x)dx \right) H(Y|Z = z)dz = \int p(z)H(Y|Z = z)dz = H(Y|Z)$$

Thus $I(Z;Y) = H(Y) - H(Y|Z) = H(Y) - H(Y|X) = I(X;Y)$.

Second, we show that $I(Z;X) = I(Z;Y)$. Since $I(Z;X) = H(Z) - H(Z|X)$, $I(Z;Y) = H(Z) - H(Z|Y)$, and $H(Z|X) = 0$, $I(Z;X) = I(Z;Y)$ is equivalent to $H(Z|Y) = 0$, i.e. $Z = p(Y|X)$ is determined by the value of $Y$.

The condition is $\forall\, \mathcal{X}_s \times \mathcal{Y}_s, X \perp\!\!\!\perp Y | \{(X, Y) \in \mathcal{X}_s \times \mathcal{Y}_s\}$, which indicates that

$$\forall\, \mathcal{X}_s \times \mathcal{Y}_s, \forall\, x, y \in \mathcal{X}_s \times \mathcal{Y}_s, \frac{p(y|x)}{p(\mathcal{Y}_s|\mathcal{X}_s)} = \frac{p(y)}{p(\mathcal{Y}_s)}$$

For any $y \in \mathcal{Y}$, choose $\mathcal{Y}_s = \{y\}$, and $\mathcal{X}_s = \{x : p(x|y) > 0\}$, we have $p(y|x) = \frac{p(\mathcal{Y}_s|\mathcal{X}_s)}{p(\mathcal{Y}_s)}p(y), \forall\, x \in \mathcal{X}_s$. Thus, $\forall\, x_1, x_2$ sampled from $p(X|Y = y)$, we have $p(y|x1) = p(y|x2)$. Therefore, $p(Y = y|X)$ is fully determined given the value of $y$ and is irrelevant to the value of $X$. Then we show that $I(Z;X) = I(Z;Y)$.

It is easy to see that deterministic mapping between $X$ and $Y$ is a special case of this condition.

$\square$

Regarding the sufficient conditions for MNI being unattainable, as outlined in Lemma 6 of Gilad-Bachrach et al. [8], when variables $X, Y$ have full support, i.e. $p(x, y) > 0, \forall\, x, y$, MNI is unattainable.

# G    Differentiability of the IB Curve

**Definition 1.** *Let $V$ and $W$ be Banach spaces, $\Omega$ an open set in $V$ and $F$ a function that maps $\Omega$ into $W$. Then the Gateaux derivative of $F$ at $x \in \Omega$ in the direction $h \in V$ is defined as*

$$dF(x;h) = \lim_{\epsilon \to 0} \frac{F(x + \epsilon h) - F(x)}{\epsilon} = \frac{d}{d\epsilon} F(x + \epsilon h) \Big|_{\epsilon=0}$$

*provided that this limit exists for all $h \in V$*

In this section, we use $X$ and $Y$ to represent the two modalities, rather than $X^1$ and $X^2$ in other sections. In our setting, the domain is the Banach space corresponding to the product space representing the set of pairs of joint probability distributions i.e. $\mathcal{Z} \times \mathcal{Y}$. The functional $F = I(Z;Y) : \Omega \to \mathbb{R}$, where $\Omega$ is the product space representing the set of pairs of joint probability distributions for which this mutual information is defined. Following the Markov structure in our graphical model, we have

$$
\begin{aligned}
I(Z;Y) &= \int_{y,z} p(z,y) \log \frac{p(z,y)}{p(z)p(y)} \\
&= \int_{x,y,z} p(z|x,y)p(x|y) \log \frac{\int_x p(z|x,y)p(x|y)}{\int_x p(z|x)p(x)} \\
&= \int_{x,y,z} p(z|x)p(x|y) \log \frac{\int_x p(z|x)p(x|y)}{\int_x p(z|x)p(x)} \\
&= F(f) \text{where } f = p(z|x)
\end{aligned}
$$

**Proposition 8.** *If the MNI point exists, then the Gateaux derivative of $I(Y;Z)$ with respect to $p(z|x)$ (as used in [24]) doesn't exist at the MNI point.*

*Proof.* Denote the MNI point between $X$ and $Y$ as $p$ where $I(X;Y) = I(Z;Y) = I(Z;X)$. When the MNI point exists, the IB curve can be represented as [27, 12]:

$$
I(Z;Y) = \begin{cases} I(Z;X), \text{ if } I(Z;X) \leq H(Y) \\ H(Y), \text{ if } H(Y) < I(Z;X) \leq H(X) \end{cases}
$$

or equivalently

$$
p(z|y) = \begin{cases} p(z|x), \text{ if } I(Z;X) \leq H(Y) \\ \mathbf{1}_{y=g(z)}, \text{ if } H(Y) < I(Z;X) \leq H(X) \end{cases}
$$

Based on this characterization, we know that there exist a direction $h$ such that on perturbing $p(z|x)$ along this directions, we either have $I(Z;Y) = I(Z;X) < H(Y)$ or $I(Z;Y) = I(X;Y) = H(Y) < I(Z;X)$. Without loss of generality, assume that perturbing by $\epsilon h$, where $\epsilon < 0$ corresponds to the former case and $\epsilon h$, where $\epsilon > 0$ to the latter. Consider the right directional derivative of I(Y:Z) at $p$ in the direction $h$, we have

$$d_+ F(p;h) = \lim_{\epsilon \to 0^+} \frac{F(p + \epsilon h) - F(p)}{\epsilon} = 0$$

Since we have $p(z|y) = p(z|x)$ when $p$ is perturbed in the direction by $\epsilon h$, where $\epsilon < 0$, we can consider $I(Z;Y)$ as an identity function of $I(Z;X)$. In such a case, $F(p + \epsilon h_1) = H(Y) + \epsilon h$. As a consequence, perturbing $p(z|x)$ to get $p'$ is equivalent to the resulting perturbation in $p(z|y)$ also

being $p'$. As a result, we have

$$
\begin{aligned}
d_- F(p; h) &= \lim_{\epsilon \to 0^-} \frac{F(p') - F(p)}{\epsilon} \\
&= \lim_{\epsilon \to 0^-} \frac{F(p(z|y) = p') - F(p)}{\epsilon} \\
&= \frac{d}{d\epsilon} F(p + \epsilon h) \Big|_{\epsilon = 0} \\
&= \frac{d}{d\epsilon} \int_{z,y} (p(z|y) + \epsilon h) p(y) \log \frac{p(z|y) + \epsilon h}{\int_y p(z|y)p(y)} \Big|_{\epsilon = 0} \\
&= \int_{z,y} hp(y) \log \frac{p(z|y) + \epsilon h}{\int_y p(z|y)p(y)} + (p(z|y) + \epsilon h)p(y) \Big( \frac{h}{p(z|y) + \epsilon h} - \frac{1}{p(z)} \Big) \Big|_{\epsilon = 0} \\
&= h \neq 0
\end{aligned}
$$

Clearly, the left and right limits exist but aren't equal, hence proving the non-differentiability.

$\square$

# H   Tractable Training Objectives

Four terms are involved in DISENTANGLEDSSL, including maximizing the mutual information term $I(Z^1; X^2)$ and minimizing the conditional mutual information term $I(Z^1; X^1|X^2)$ for the shared representations in step 1, as well as maximizing the joint mutual information term $I(Z^1, \hat{Z}_c^{2*}; X^1)$ and minimizing the mutual information term $I(Z^1; \hat{Z}_c^{1*})$ for the modality-specific representations in step 2. We introduce detailed formulations of the tractable training objectives for each of the four terms in this section.

For the inferred shared representations, we model the distributions $\hat{Z}_c^1 \sim p(\cdot|X^1)$ and $\hat{Z}_c^2 \sim p(\cdot|X^2)$ with neural network encoders. Following the common practice in Radford et al. [25], we use the InfoNCE objective [23] as an estimation of the mutual information term $I(Z^1; X^2)$ in $L_c^1$.

$$
L_{\text{InfoNCE}}^c = \mathbb{E}_{z^1, z^{2+}, \{z_i^{2-}\}_{i=1}^N} \left[ -\log \frac{\exp(z^{1\top} z^{2+}/\tau)}{\exp(z^{1\top} z^{2+}/\tau) + \sum_{i=1}^N \exp(z^{1\top} z_i^{2-}/\tau)} \right]
$$

where $\tau$ is the temperature hyperparameter, $z^{2+}$ is the representation of the positive sample corresponding to the joint distribution $p(X^1, X^2)$, and $\{z_i^{2-}\}_{i=1}^N$ are representations of $N$ negative samples from the marginal distribution $p(X^2)$.

We implement the conditional mutual information term $I(Z^1; X^1|X^2)$ in $L_c^1$ using an upper bound developed in Federici et al. [6], i.e. $I(Z^1; X^1|X^2) \leq D_{\text{KL}}(p(Z^1|X^1)||p(Z^2|X^2))$. While the conditional distributions of representations are modeled as the Gaussian distribution in Federici et al. [6], we instead use the von Mises-Fisher (vMF) distribution for $p(Z^1|X^1)$ and $p(Z^2|X^2)$ to better align with the InfoNCE objective where the representations lie on the sphere space. Specifically, $\hat{Z}_c^1 \sim \text{vMF}(\mu(X^1), \kappa), \hat{Z}_c^2 \sim \text{vMF}(\mu(X^2), \kappa)$ where $\kappa$ is a hyperparameter controlling for the uncertainty of the representations. Leveraging the formulation of the KL divergence between two vMF distributions, the training objective of $L_c^1$ is to maximize:

$$
L_c^1 = -L_{\text{InfoNCE}}^c + \beta \cdot \mathbb{E}_{x^1, x^2} \left[ \mu(x^1)^\top \mu(x^2) \right]
$$

Note that this objective establish connections with the alignment versus uniformity framework discussed in Wang and Isola [37], where the conditional information bottleneck constraint corresponds to a higher weight on the alignment term, in contrast to the uniformity term.

The inferred modality-specific representations are encoded as functions $\hat{Z}_s^1 \sim p(\cdot|X^1, \hat{Z}_c^1)$ and $\hat{Z}_s^2 \sim p(\cdot|X^2, \hat{Z}_c^2)$ with deterministic encoders, that takes both data observations and the shared representations learned in step 1 as input to account for the dependence structure illustrated in Figure

2. The term $I(Z^1, \hat{Z}_c^{2*}; X^1)$ in $L_s^1$ is optimized with the InfoNCE loss, where random augmentations of the data $X^1$ form the two views. Denote the concatenation of $Z^1$ and its corresponding $\hat{Z}_c^{2*}$ as $\tilde{Z}^1$, the InfoNCE objective has the following formula:

$$L_{\text{InfoNCE}}^s = \mathbb{E}_{\tilde{z}^1, \tilde{z}^{1+}, \{\tilde{z}_i^{1-}\}_{i=1}^N} \left[ -\log \frac{\exp(\tilde{z}^{1\top} \tilde{z}^{1+}/\tau)}{\exp(\tilde{z}^{1\top} \tilde{z}^{1+}/\tau) + \sum_{i=1}^N \exp(\tilde{z}^{1\top} \tilde{z}_i^{1-}/\tau)} \right]$$

where $\tilde{z}^{1+}$ is the representation of the positive sample corresponding to the augmented view of $X^1$, and $\{z_i^{1-}\}_{i=1}^N$ are representations of $N$ negative samples from the marginal distribution $p(X^1)$.

For the mutual information term between representations, $I(Z^1; \hat{Z}_c^{1*})$ in $L_s^1$, we implement it as an orthogonal loss to encourage the marginal independence between the shared and modality-specific representations, where the marginal distribution is approximated across a training batch.

$$L_{\text{orthogonal}} = ||[Z_i^1]_{i=1}^B{}^\top \cdot [\hat{Z}_{ci}^{1*}]_{i=1}^B||_F$$

where $B$ is the batch size, $[Z_i^1]_{i=1}^B$ and $[\hat{Z}_{ci}^{1*}]_{i=1}^B$ are the concatenations of all the representations in a mini-batch, and $|| \cdot ||_F$ is the Frobenius norm of the pairwise cosine similarities between each latent dimensions. The training objective of $L_s^1$ in step 2 is to maximize:

$$L_s^1 = -L_{\text{InfoNCE}}^s - \lambda \cdot L_{\text{orthogonal}}$$

# I   Experimental Details and Additional Results

Each experiment was conducted on 1 NVIDIA RTX A5000 GPU, each with 24GB of accelerator RAM. All experiments were implemented using the PyTorch deep learning framework.

## I.1   Simulation Study

**Synthetic data generation.** We generate synthetic data based on the graphical model in Figure 2. We sample $d$-dimensional true latents $Z_s^1$, $Z_s^2$, and $Z_c$ independently from $\mathcal{N}(\mu_d, \Sigma_d^2)$. Using fixed transformations $T_1$ and $T_2$, we create $X^1 = T_1 \cdot [Z_s^1, Z_c]$ and $X^2 = T_2 \cdot [Z_s^2, Z_c]$. To simulate unattainable MNI, we add Gaussian noise to ensure the distribution has full support. Binary labels $Y_s^1$, $Y_s^2$, and $Y_c$ are constructed from the true latents and used to evaluate the information content of learned representations via linear probing accuracy. Denote $\hat{Z}_c$ as the combination of the learned shared representations of $X^1$ and $X^2$, i.e. $\hat{Z}_c = [\hat{Z}_c^1, \hat{Z}_c^2]$. Ideally, $\hat{Z}_c$ should achieve high accuracy on $Y_c$ and low on $Y_s^1$ and $Y_s^2$, while the modality-specific representations $\hat{Z}_s^1$ and $\hat{Z}_s^2$ should show the opposite pattern. Additional details on experimental settings are elaborated below.

**Experimental details.** We generate synthetic data $X^1$ and $X^2$ based on the graphical model in Figure 2, with the dimensionality of 100 and dataset size being 90,000. To be specific, we sample 50-dimensional true latents $Z_s^1$, $Z_s^2$, and $Z_c$ independently from $\mathcal{N}(\mathbf{0}_{50}, 0.5 \times \mathbf{I}_{50})$. Then we sample the transformation weights $T_1$ and $T_2$ from uniform distribution Uniform$(-1, 1)$, and generate $X^1 = T_1 \cdot [Z_s^1, Z_c]$ and $X^2 = T_2 \cdot [Z_s^2, Z_c]$. We randomly split $80\%$ data into the training set and the rest into the test set. To simulate unattainable MNI, we add Gaussian noise and random dropout during training to ensure the distribution has full support. We use a 3-layer multi-layer perceptron (MLP) with a hidden dimension of 512 as encoders for all methods. For DMVAE, we employ MLPs with the same architecture as decoders. We report the average linear probing accuracy on the test set over 3 random seeds.

We run all the methods on the synthetic data with combinations of different hyperparameter values. For DISENTANGLEDSSL, we use $\beta \in \{0.0, 0.001, 0.01, 0.1, 0.5, 1.0, 5.0, 10.0, 50.0, 100.0, 300.0, 500.0, 1000.0\}$ and $\lambda \in \{0.0, 0.001, 0.01, 0.1, 1.0, 10.0, 100.0\}$.

For JointOpt, hyperparameter $a$ controls the joint mutual information terms $I(\hat{Z}_s^1, \hat{Z}_c^2; X^1)$ and $I(\hat{Z}_s^2, \hat{Z}_c^1; X^2)$, and $\lambda$ adjusts the mutual information term between representations, i.e.

$$\hat{Z}_c^{1*}, \hat{Z}_c^{2*}, \hat{Z}_s^{1*}, \hat{Z}_s^{2*} = \underset{\hat{Z}_c^1, \hat{Z}_c^2, \hat{Z}_s^1, \hat{Z}_s^2}{\arg\max} \ I(\hat{Z}_c^1; X^2) + I(\hat{Z}_c^2; X^1) + a \cdot (I(\hat{Z}_s^1, \hat{Z}_c^2; X^1) + I(\hat{Z}_s^2, \hat{Z}_c^1; X^2))$$
$$- \lambda \cdot (I(Z_c^1; Z_s^1) + I(Z_c^2; Z_s^2))$$

We use $a \in \{0.01, 0.1, 1.0, 10.0, 100.0, 1000.0\}$ and $\lambda \in \{0.0, 0.001, 0.01, 0.1, 1.0, 10.0, 100.0\}$.

For FOCAL, we tune the hyperparameters $a$ and $\lambda$, defined similarly to JointOpt, where $a$ controls the terms $I(\hat{Z}_s^1; X^1)$ and $I(\hat{Z}_s^2; X^2)$ and $\lambda$ adjusts the orthogonal loss between shared and specific representations. We use the same set of $a$ and $\lambda$ as that in JointOpt.

For DMVAE, we tune $\lambda$ which denotes the weight of the KL divergence term in contrast to the reconstruction loss. We use $\lambda \in \{10^{-7}, 10^{-6}, 10^{-5}, 10^{-4}, 10^{-3}, 10^{-2}\}$.

**Results overview.** We assess the performance of the learned shared and modality-specific representations for different values of $\beta$ and $\lambda$, as shown in Figure 6. For comparison, we also evaluate JointOpt, DMVAE, and FOCAL across different hyperparameter settings. Specifically, for JointOpt, we vary both $a$ and $\lambda$, where $a$ controls the joint mutual information terms $I(\hat{Z}_s^1, \hat{Z}_c^2; X^1)$ and $I(\hat{Z}_s^2, \hat{Z}_c^1; X^2)$, and $\lambda$ adjusts the mutual information term between representations, similar to DISENTANGLEDSSL.

Figure 6a illustrates the performance of shared representation $\hat{Z}_c$ learned by DISENTANGLEDSSL across different values of $\beta$. For lower values of $\beta$, $\hat{Z}_c$ captures both shared and specific features, as indicated by linear probing accuracy on $Y_s^1$, $Y_s^2$, and $Y_c$ exceeding 0.5. As $\beta$ increases, all accuracies decrease, reflecting the trade-off between expressivity and redundancy controlled by $\delta_c$ when MNI is unattainable. This trend aligns well with Figure 3 and Proposition 2.

Given the shared representations $\hat{Z}_c^1$ and $\hat{Z}_c^2$ learned in step 1 for a fixed $\beta$, we then learn the corresponding modality-specific representations $\hat{Z}_s^1$ and $\hat{Z}_s^2$ with varying $\lambda$. Figure 6b shows the performance of DISENTANGLEDSSL in contrast to other baseline methods, where dots are connected according to a descending order of their corresponding $\lambda$ values[5]. The ideal modality-specific representation $\hat{Z}_s^1$ should maximize unique information from $X^1$, shown by high accuracy on $Y_s^1$, while minimizing shared information with $X^2$, indicated by low accuracy on $Y_c$. Therefore, a bottom-right point is preferred in Figure 6b. As illustrated in Figure 6b, DISENTANGLEDSSL outperforms all other methods across various hyperparameter settings, especially JointOpt, demonstrating the effectiveness of the stepwise optimization procedure. Results on $\hat{Z}_s^2$ are provided in Figure 8.

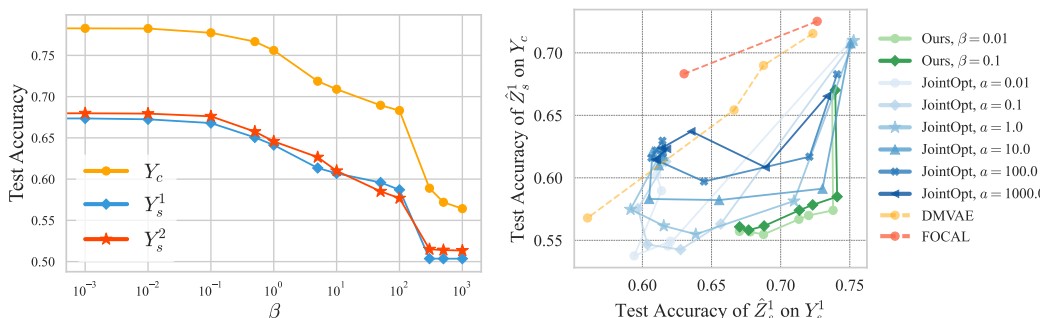

(a) Performance of shared representation $\hat{Z}_c$ learned with different values of $\beta$.

(b) Performance of modality-specific representation $\hat{Z}_s^1$ learned with different $\beta$ and $\lambda$, in comparison to baselines.

Figure 6: Simulation study results.

**Additional results.** We provide a comparison of the performance of the shared representation with baseline methods in Figure 7. Denote $\hat{Z}_c$ as the concatenation of the learned shared representations of $X^1$ and $X^2$, i.e. $\hat{Z}_c = [\hat{Z}_c^1, \hat{Z}_c^2]$. The ideal $\hat{Z}_c$ should maximize the shared information between $X^1$ and $X^2$, shown by high accuracy on $Y_c$, while minimizing unique information of $X^1$ and $X^2$, indicated by low accuracy on $Y_s^1$ and $Y_s^2$. Therefore, a top-left point is preferred in Figure 7.

As shown in Figure 7, DISENTANGLEDSSL consistently outperforms other methods across various hyperparameter settings, showcasing its ability to effectively capture shared information. The only exception occurs in Figure 7a when $\beta$ is set very high and the accuracy on $Y_s^1$ to drop to around 0.50

---

[5]For FOCAL, we tune the hyperparameters $a$ and $\lambda$, defined similarly to JointOpt, where $a$ controls the terms $I(\hat{Z}_s^1; X^1)$ and $I(\hat{Z}_s^2; X^2)$ and $\lambda$ adjusts the orthogonal loss between shared and specific representations. For DMVAE, we tune $\lambda$ which denotes the weight of the KL divergence term. We show the best-performing results of FOCAL and DMVAE across hyperparameters in Figure 6b, with full results available in Figure 8.

(equivalent to random guessing, indicating no information of $\hat{Z}_c$ on modality-specific features). In this scenario, DMVAE performs better. This happens because high values of $\beta$ cause the decoder-free contrastive objectives to collapse, with most representations converging to nearly the same point, a commonly-known issue in contrastive self-supervised learning [23].

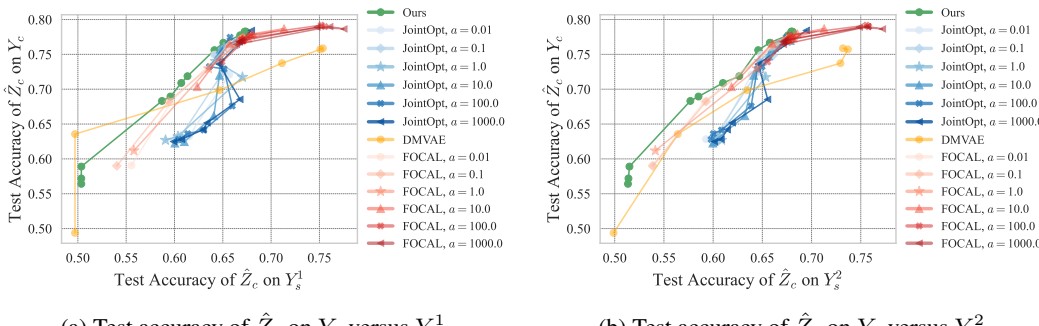

(a) Test accuracy of $\hat{Z}_c$ on $Y_c$ versus $Y_s^1$.

(b) Test accuracy of $\hat{Z}_c$ on $Y_c$ versus $Y_s^2$.

Figure 7: Performance of shared representation $\hat{Z}_c$ for different models.

For the modality-specific representations, as a supplement to Figure 6b, we provide the complete results for both representations $\hat{Z}_s^1$ and $\hat{Z}_s^2$ on a full set of hyperparameters in Figure 8. DISENTAN-GLEDSSL outperforms all other methods across various hyperparameter settings.

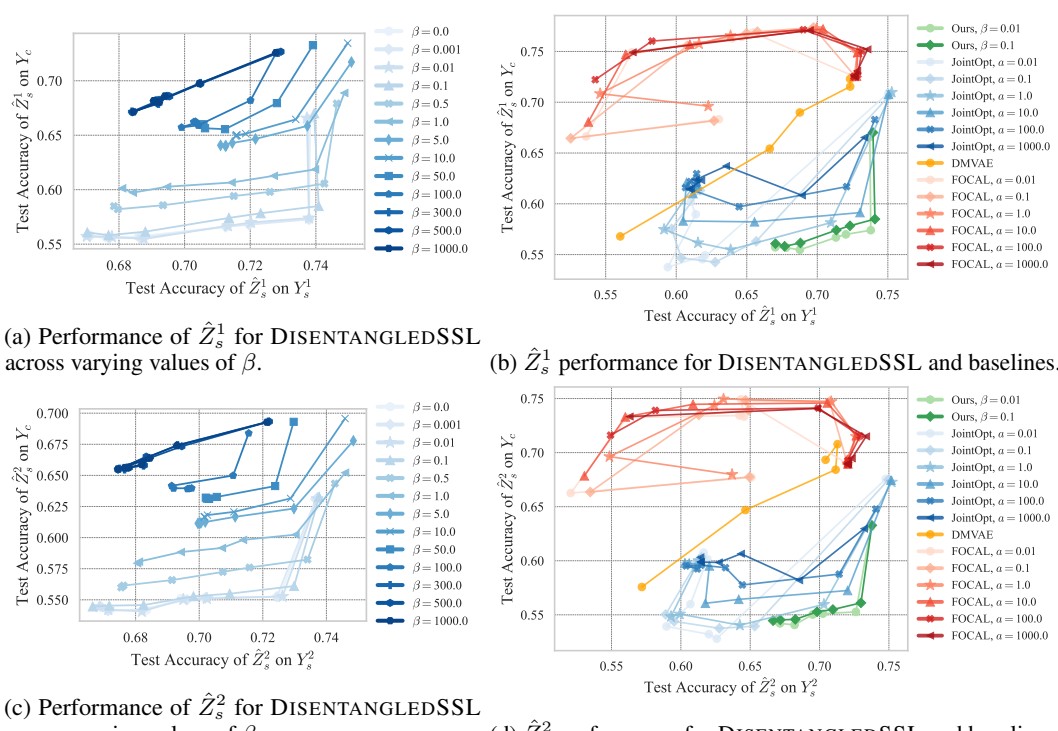

(a) Performance of $\hat{Z}_s^1$ for DISENTANGLEDSSL across varying values of $\beta$.

(b) $\hat{Z}_s^1$ performance for DISENTANGLEDSSL and baselines.

(c) Performance of $\hat{Z}_s^2$ for DISENTANGLEDSSL across varying values of $\beta$.

(d) $\hat{Z}_s^2$ performance for DISENTANGLEDSSL and baselines.

Figure 8: Performance of modality-specific representation $\hat{Z}_s^1$ and $\hat{Z}_s^2$ for different models.

**Analysis for different levels of entanglement.** We further examine the trade-off between expressivity and redundancy for the learned shared representations on synthetic data with varying levels of multimodal entanglement. In this scenario, some dimensions align across modalities with MNI attainable, while others remain entangled with MNI unattainable. Specifically, we split the 50-dimensional true shared latent $Z_c$ into two parts: the first 35 dimensions, denoted as $Z_c^{\mathrm{mix}}$, and the last 15 dimensions, denoted as $Z_c^{\mathrm{pure}}$. We then generate 100-dimensional observations $X^1$ and $X^2$, where the first 85 dimensions are generated under the same procedure as before, i.e. $X_{\mathrm{mix}}^1 = T_1 \cdot [Z_s^1, Z_c^{\mathrm{mix}}]$

and $X^2_{\text{mix}} = T_2 \cdot [Z^2_s, Z^{\text{mix}}_c]$ followed by adding Gaussian noise and random dropout, while the last 15 dimensions are directly $Z^{\text{pure}}_c$.

We present the results on the synthetic data with a mixed level of entanglement in Figure 9. Figure 9a shows the test accuracy of $\hat{Z}^1_c$ on $Y_c$, $Y^1_s$, and $Y^2_s$ on the left axis, alongside the MLP weight ratio on $X^1$ and $X^2$ on the right axis, for varying values of $\beta$. To compute the MLP weight ratio, we first extract the diagonal of the inner product of MLP encoder's first layer weight matrix, then calculate the ratio between the average of the last 15 "pure" dimensions and the first 85 "mixed" dimensions. This ratio indicates how much attention the encoder gives to the "pure" versus "mixed" dimensions, with higher values signifying greater focus on the "pure" dimensions. Figure 9b shows the corresponding test accuracy on $Y_c$ versus $Y^2_s$ in line plot.

Using such data with mixed entanglement levels, DISENTANGLEDSSL demonstrates a clear pattern where the learned information plateaus at certain $\beta$ values. As illustrated in Figure 9, the MLP weight ratio initially rises sharply from around 20 to nearly 80, then drops to 1 when $\beta$ becomes very large, indicating the collapse of the learned representations. With a large value of $\beta$ (while before the model collapses, i.e. $\beta \approx 10$), the encoders focus mainly on the "pure" dimensions. This is because a stronger information bottleneck constraint discourages the extraction of shared components from the "mixed" dimensions, which inevitably include modality-specific information due to unattainable MNI, and favors the "pure" shared components with no extra cost.

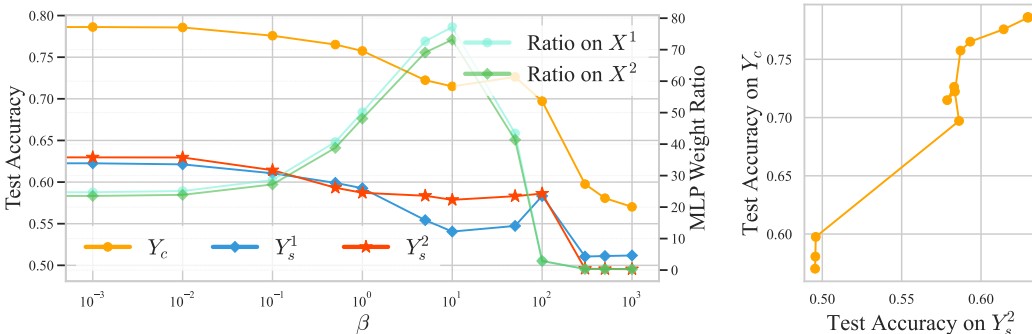

(a) Test accuracy of $\hat{Z}^1_c$ and the MLP weight ratio with varying $\beta$.  (b) Test accuracy on $Y_c$ versus $Y^2_s$.

Figure 9: Performance of $\hat{Z}_c$ on synthetic data with a mixed level of entanglement.

## I.2  MultiBench.

**Experimental details.** We follow the same dataset splitting, and utilize the same encoder architecture and pre-extracted features as [17]. All models use representations (or concatenation of them, for FactorCL-proj, FOCAL, JointOpt and DISENTANGLEDSSL) with the same dimensionality. We set the latent dimension to 300 across all datasets, except for MUSTARD, where it is set to 100 due to the smaller dataset size, and for MIMIC, where we use 360 to align with the output dimension of the GRU encoders. For FactorCL, we use their default hyperparameter settings. For other methods, hyperparameters are tuned based on validation set performance. For DISENTANGLEDSSL, we use $\beta = 1.0$ and $\lambda = 10^{-3}$ for all datasets, except for MOSI where $\beta = 0.01$. For FOCAL and JointOpt, we use $a = 1$ and $\lambda = 10^{-3}$ across all datasets. We report the mean and standard deviation of the linear probing accuracy on prediction labels from the test set over 3 random seeds.

**Additional results.** As a complement to Table 1, we present results of FOCAL and JointOpt for the learned shared and modality-specific representations, both combined (i.e. "both") and separately in Table 4. Although they exhibit a similar trend as DISENTANGLEDSSL, where shared representations are more important for MOSI and specific representations are crucial for MUSTARD, the distinction is less pronounced. Moreover, the overall performance is inferior to that of DISENTANGLEDSSL, indicating that DISENTANGLEDSSL achieves a better balance between coverage and disentanglement in the learned representations.

## I.3  High-Content Drug Screening

**Experimental details.** Following the setup in Wang et al. [36], we utilize Mol2vec [11] to featurize the molecular structures into 300-dimensional feature vectors. For both molecular structures and

Table 4: Prediction accuracy (%) of the representations learned by different methods on MultiBench datasets and standard deviations over 3 random seeds.

| Dataset | MIMIC | MOSEI | MOSI | UR-FUNNY | MUSTARD |
|---|---|---|---|---|---|
| CLIP | 64.97 (0.60) | 76.87 (0.45) | 64.24 (0.88) | 62.73 (0.92) | 56.04 (4.19) |
| FactorCL-emb | 65.25 (0.45) | 71.80 (0.64) | 62.97 (0.81) | 63.29 (2.07) | 56.76 (4.66) |
| FactorCL-proj | 59.43 (1.70) | 74.61 (1.65) | 56.02 (1.26) | 61.25 (0.47) | 55.80 (2.18) |
| FOCAL (shared) | 62.69(1.46) | 75.93(0.23) | 62.10(0.44) | 62.38(0.58) | 54.83(3.02) |
| FOCAL (specific) | 62.13(1.49) | 75.41(0.54) | 61.61(1.27) | 63.17(0.96) | 58.21(2.21) |
| FOCAL (both) | 64.42 (0.34) | 76.77 (0.51) | 63.65 (1.09) | 62.98 (1.52) | 54.35 (0.00) |
| JointOpt (shared) | 63.01(0.59) | 76.69(0.28) | 65.02(1.96) | 62.51(1.02) | 54.83(4.82) |
| JointOpt (specific) | 65.81(0.49) | 74.40(0.94) | 53.89 (0.80) | 62.13(0.69) | 57.73(4.12) |
| JointOpt (both) | 66.11 (0.64) | 76.71 (0.14) | 64.24 (1.75) | 63.58 (1.45) | 56.52 (2.61) |
| DISENTANGLEDSSL (shared) | 63.16 (0.48) | 76.94 (0.22) | **65.16** (0.81) | 64.14 (1.53) | 54.11 (1.51) |
| DISENTANGLEDSSL (specific) | 65.73 (0.09) | 75.99 (0.60) | 51.70 (0.72) | 60.27 (1.28) | **61.60** (2.61) |
| DISENTANGLEDSSL (both) | **66.44** (0.31) | **77.45** (0.06) | 65.11 (0.80) | **64.24** (1.54) | 56.52 (2.18) |

phenotypes, we employ 3-layer MLP encoders with a hidden dimension of 2560. The dimensionality for shared and modality-specific representations is set to 32 across all methods and datasets. We tune $\beta$ based on validation set performance, with $\beta = 5.0$ for RXRX19a and $\beta = 1.0$ for LINCS, and set $\lambda = 0.01$ for both datasets. For FOCAL and JointOpt, we set $a = 1.0$ and $\lambda = 0.01$. For DMVAE, we set the coefficient of the KL divergence term as $10^{-5}$. In addition, as in Lee and Pavlovic [14], we introduce an InfoNCE loss for the shared representations to DMVAE in the high-content drug screening experiments to enhance its retrieving accuracy. We report the mean and standard deviation of all results on the test set over 3 random seeds.

**Counterfactual generation.** To further evaluate the learned modality-specific representations, we use different combinations of the shared and modality-specific representations to generate counterfactual samples, i.e. predicting phenotype of a drug on a different cell with its molecule shared latent but the phenotype specific latent of a different cell perturbed by other drugs. Since the factual observations of counterfactual generation are unavailable, we measure the performance distributional-wise, introducing the difference in Frechet distance (Diff-FD) as a metric, i.e. Diff-FD-c measuring the gain of using the correct batch of molecules, thus the information of the shared latent; Diff-FD-s measures the gain of using the correct batch of cells, thus the information of the specific latent.

To be specific, we train decoders with the factual combination of the shared latents from molecular structures and the modality-specific latents from phenotypes on the training set. We use "val recon" to denote the samples generated using the representations attained from validation set molecules and validation set phenotypes as input. Similarly, "test recon" refers to the samples generated using test set molecules paired with test set phenotypes, while "counterfactual" represents those generated using test set molecules paired with validation set phenotypes. Formally, Diff-FD-c is defined as FD(val recon; test recon) - FD(counterfactual; test recon), and Diff-FD-s as FD(val recon; test recon) - FD(counterfactual; val recon). Note that a non-informative specific latent can lead to very high Diff-FD-c, and an overly informative modality-specific latent can lead to very high Diff-FD-s, thus we take both metrics into consideration together.

Table 5: Results on counterfactual generation.

| Dataset | RXRX19a | | LINCS | |
|---|---|---|---|---|
| Metric | Diff-FD-c | Diff-FD-s | Diff-FD-c | Diff-FD-s |
| FOCAL | 15.30(0.85) | **22.89**(0.28) | 0.246(0.039) | **0.765**(0.027) |
| JointOpt | 20.34(0.47) | 9.40(0.39) | 0.248(0.029) | 0.333(0.050) |
| DISENTANGLEDSSL | **20.92**(0.71) | 11.36(0.36) | **0.277**(0.031) | 0.401(0.085) |

As shown in Table 5, DISENTANGLEDSSL outperforms JointOpt in both metrics and surpasses FOCAL in Diff-FD-s. While FOCAL shows a high Diff-FD-s, it learns overly informative modality-specific latents, which contains significant shared information, as highlighted in the simulation study results (i.e. the modality-specific representations of FOCAL have high accuracy on $Y_c$ in Figure 8), leading to a low value of Diff-FD-c. In contrast, DISENTANGLEDSSL demonstrate better effectiveness in capturing both shared and modality-specific information, while maximizing the separation between them, as indicated by its high values in both metrics.

