# OpenReview forum: "An Information Criterion for Controlled Disentanglement of Multimodal Data"
_NeurIPS.cc/2024/Workshop/UniReps — UniReps_

### Official Review · Reviewer_eqUL · 2024-10-04
**Good investgation of dientanglement of multimodal data**

**Rating:** 7
**Confidence:** 3

**Review:**

# Summary
This paper proposed an interesting view to handle multimodal representation learning that disentangles shared and modality-specific information using an information-theoretic framework. It introduces a step-by-step optimization strategy to learn these representations effectively, even when the Minimum Necessary Information (MNI) point is not attainable.

# Strength
1. The proposed self-supervised method operates effectively even when the Minimum Necessary Information (MNI) point is not attainable, addressing a common limitation in multimodal learning. It is interesting to show the comparison of the IB curves of non-MNI and MNI.
2. The approach is grounded in information theory, utilizing the Information Bottleneck (IB) principle and its variants. By defining clear information-theoretic criteria, it ensures that both shared and specific representations are informative and disentangled. The paper provides rigorous analysis and formal proofs (in the appendices).
3. Step-by-step optimization makes sense to me. Adopting a sequential optimization strategy to disentangle shared and modality-specific information is more appropriate for such a complex task.

# Weakness
1. Limited discussion on scalability. Although the paper demonstrates empirical success on datasets like MultiBench and high-content drug screening, there is limited discussion on how well the model scales with increasing data complexity or dimensionality.
2. The experimental results seem to not be promising compared with other baselines.

---

### Official Review · Reviewer_be3q · 2024-10-05
**Technically solid paper with conclusive results**

**Rating:** 7
**Confidence:** 3

**Review:**

### **Summary**

This paper addresses the challenge of disentanglement between modalities for self-supervised learning regimes by introducing DisentangledSSL that learns disentangled representations across modalities. DSSL outperforms on V&L prediction and phenotyping tasks in biology. The paper builds on the concept of minimum necessary information and leverages this information-theoretic metric to ensure disentanglement between modality-specific and shared representations.

### Strengths

- The motivation of the paper is clear and  concise
- The method of the paper is well-crafted, even if sometimes disjoint with the parts that were moved to the appendix.
- The results are conclusive and support the claims made in the paper
- Assuming that runtime of the two-step training process is similar to the benchmarks, this would be a feasible alternative to CLIP and other contrastive SSL methods across domains.

### Areas for Development/Questions

- Reading the paper, it is unclear what the “true latent” (L42) is. Is this supposed to be a conceptual explanation?
- The method section is relatively clear, although in parts disjoint. It would be helpful to indicate which parts are just establishing information and which are a novelty of the paper to further improve the clarity of the contributions. Additionally, it would strengthen the paper to link the appendices in this section for further explanations.
- Making the figure captions self-contained and linked to the main body would improve clarity.
- It’s very obvious that some sections from a full paper submission were moved down to the abstract. However, this needs to be done with care - introducing important abbreviations like IB (L303) should be in the main body. This reduces the readability of this paper.
- How does the two-step learning process compare to the benchmark in terms of wall time?

---

### Official Review · Reviewer_ninu · 2024-10-06

**Rating:** 9
**Confidence:** 3

**Review:**

## Summary
This paper introduces DISENTANGLEDSSL, a self-supervised learning method designed to achieve controlled disentanglement of multimodal data. The method incorporates information-theoretic criteria and a two-step optimization strategy to learn disentangled representations even when the Minimum Necessary Information (MNI) point is unattainable. Empirical results on multiple datasets demonstrate that DISENTANGLEDSSL outperforms existing baselines in terms of disentanglement quality and downstream task performance.

## Strengths
1. The proposed method provides a unique approach to disentangling shared and modality-specific information in multimodal data.
2. The paper offers a theoretical foundation by leveraging information theory, particularly the information bottleneck framework.
3. Extensive experiments on several datasets shows its effectiveness.

## Weakness
1. It would be better to add feature distribution visualizations, such as TSNE.

---

### Decision · Program_Chairs · 2024-10-10

**Decision:**

Accept (Oral)

**Comment:**

In light of the positive reviewers' feedback and relevancy of the submission, we are pleased to accept this paper for presentation at UniReps 2024. We kindly ask the authors to incorporate the reviewers' suggestions and feedback in the final camera-ready version of the manuscript.